# Tacrine-Coumarin Derivatives as Topoisomerase Inhibitors with Antitumor Effects on A549 Human Lung Carcinoma Cancer Cell Lines

**DOI:** 10.3390/molecules26041133

**Published:** 2021-02-20

**Authors:** Eva Konkoľová, Monika Hudáčová, Slávka Hamuľaková, Rastislav Jendželovský, Jana Vargová, Juraj Ševc, Peter Fedoročko, Mária Kožurková

**Affiliations:** 1Department of Biochemistry, Institute of Chemistry, Faculty of Science, P. J. Šafárik University in Kosice, 041 80 Košice, Slovakia; eva.konkolova@upjs.sk (E.K.); monika.hudacova@student.upjs.sk (M.H.); 2Institute of Organic Chemistry and Biochemistry AS CR, Flemingovo námestí 2, 160 00 Prague 6, Czech Republic; 3Department of Organic Chemistry, Institute of Chemistry, Faculty of Science, P. J. Šafárik University in Košice, 041 80 Košice, Slovakia; slavka.hamulakova@upjs.sk; 4Department of Cellular Biology, Institute of Biology and Ecology, Faculty of Science, P. J. Šafárik University in Košice, 041 80 Košice, Slovakia; rastislav.jendzelovsky@upjs.sk (R.J.); jana.vargova@upjs.sk (J.V.); juraj.sevc@upjs.sk (J.Š.); peter.fedorocko@upjs.sk (P.F.); 5Biomedical Research Center, University Hospital Hradec Kralove, 500 05 Hradec Kralove, Czech Republic

**Keywords:** tacrine-coumarin derivatives, DNA, topoisomerases I, II, cytotoxicity, lung carcinoma cells, A549

## Abstract

A549 human lung carcinoma cell lines were treated with a series of new drugs with both tacrine and coumarin pharmacophores (derivatives **1a**–**2c**) in order to test the compounds’ ability to inhibit both cancer cell growth and topoisomerase I and II activity. The ability of human topoisomerase I (*h*TOPI) and II to relax supercoiled plasmid DNA in the presence of various concentrations of the tacrine-coumarin hybrid molecules was studied with agarose gel electrophoresis. The biological activities of the derivatives were studied using MTT assays, clonogenic assays, cell cycle analysis and quantification of cell number and viability. The content and localization of the derivatives in the cells were analysed using flow cytometry and confocal microscopy. All of the studied compounds were found to have inhibited topoisomerase I activity completely. The effect of the tacrine-coumarin hybrid compounds on cancer cells is likely to be dependent on the length of the chain between the tacrine and coumarin moieties (**1c**, **1d** = tacrine-(CH_2_)_8–9_-coumarin). The most active of the tested compounds, derivatives **1c** and **1d**, both display longer chains.

## 1. Introduction

Coumarins have attracted a great deal of attention due to the wide range of their biological properties [1,2,3,4]. Recent research has focused attention on the anticancer activity of coumarin and coumarin-derived compounds due to their high level of biological activity and low toxicity [5,6,7]. Coumarins are commonly used in the treatment of prostate cancer, colon, renal cell carcinoma and leukemia in particular [8,9,10]. Further research has also led to irusostat (a potent coumarin-based irreversible inhibitor) compounds entering clinical trials for possible future use in the treatment of breast cancer [11,12,13]. Lung cancer is one of the most commonly diagnosed malignant tumors and is the leading cause of cancer death throughout the world. The currently available therapies in the treatment of advanced lung cancer, primarily radiotherapy and chemotherapy, are still inadequate. While highly effective FDA-approved drugs such as, e.g., efitinib, erlotinib, and bevacizumab are now available for targeted therapy/chemotherapy, these drugs can cause side effects [14]. Therefore, there is an urgent need for the development of novel drugs for treating this disease. A549 human lung carcinoma cells are a well characterized cellular model for this purpose [15,16].

Different mechanisms are thought to be responsible for the anticancer activity of coumarins, including the blocking of the cell cycle, the induction of cell apoptosis, the modulation of the estrogen receptor, or the inhibition of DNA-associated enzymes such as telomerase and topoisomerase (TOP). Topoisomerase enzymes play an important role in DNA metabolism, and the search for novel enzyme inhibitors is an important target in the development of new anticancer drugs [17,18].

The relevance and significance of these compounds is obvious, and the agents have attracted considerable attention through the development of novel biologically active molecules. The approach is based on the highly effective combination principle of drug design and involves the coupling of coumarins with other bioactive molecules [19,20,21,22]. The activity of tacrine (9-amino-1,2,3,4-tetrahydroacridine) in neurological disorders such as Alzheimer’s disease is now well established. Numerous studies have confirmed that the drug is an effective inhibitor of acetylcholinesterase [22,23,24,25] and it has also been reported that it is not clastogenic in mammalian cells [26]. Tacrine is a relatively weak catalytic inhibitor of TOPII (in comparison with 9-aminoacridine), which has been found to inhibit topoisomerase and DNA synthesis, thereby resulting in mitochondrial DNA depletion and apoptosis [27,28,29]. Hybrid molecules, formed by the combination of two or more pharmacophores, is an emerging concept in the field of medicinal chemistry and drug discovery that has attracted substantial attraction in the past few years [30].

The aim of this study is to show that these structurally novel tacrine-coumarin compounds, derivatives **1a**–**1d** and **2a**–**2c**, may exhibit anticancer properties and also to examine the antiproliferative and topoisomerase activities of the derivatives in more detail.

## 2. Results

### 2.1. Topoisomerase Relaxation Assay

The ability of human topoisomerase I (*h*TOPI) to relax supercoiled plasmid DNA in the presence of various concentrations of tacrine-coumarin hybrid molecules was studied with agarose gel electrophoresis and the results are shown in Figure 1. The results clearly show that supercoiled plasmid DNA (line *p*BR322) was fully relaxed under normal conditions with *h*TOPI (*h*TOPI line + *p*BR322). However, relaxation induced by *h*TOPI was inhibited when the concentration of the studied compounds (lines **1a**–**2c**) was gradually increased. All of the studied compounds were found to have caused partial inhibition of topoisomerase activity at a concentration of 30 × 10^−6^ M and complete inhibition was detected at a concentration of 60 × 10^−6^ M.

In order to evaluate whether the compounds can also inhibit topoisomerase IIα activity, the decatenation of catenated plasmid DNA was performed in the presence of the compounds. However, no significant inhibitory effect was detected, even at the highest concentration of 100 × 10^−6^ M concentration (data not shown) and therefore we suggest that compounds **1a**–**2c** are unable to inhibit the topoisomerase IIα enzyme.

### 2.2. Intracellular Localization and Cytotoxicity Assays

Flow cytometric analysis of the content of the derivatives present in A549 cells revealed the cumulative fluorescence of derivatives **1b**–**1d** and **2b** from the green (FL-1) to the red (FL-3) channel (Figure 2). Compound **1c** was found to display the highest level of fluorescence. The presence of compounds **1a**–**2c** in A549 human adherent lung carcinoma cells were analysed by observing the fluorescence of the compounds in the green channel (*Ex* = 488 nm, *Em* = 510–560 nm). This analysis was performed in order to detect the accumulation of the compounds within the cells by exploiting their natural fluorescence. This allowed us to correlate the accumulation of the derivatives with their observed effects on the cellular parameters. The accumulation of the compounds was then investigated in more detail with respect to their specific intracellular distribution using confocal microscopy.

According to our results (Figure 3), compound **1d** displayed the highest rate of detection in cells, with compounds **1c** and **1b** also showing weaker levels of detection. In other samples, the fluorescence of the derivatives could not be distinguished from the autofluorescence of the cancer cells. At the cellular level, the analyzed compounds were distributed in the cytoplasm with no interference with the cell nucleus. Based on mitochondrial staining and the overall distribution of the signal, we could not confirm the accumulation of the derivatives in the mitochondria or in the other organelles or membranes (data not shown).

### 2.3. MTT Assay

The ability of the studied compounds to inhibit the metabolic activity of A549 cancer cell lines was determined using an MTT assay. Results were obtained from three independent experiments and each experiment was carried out in triplicate. As is evident from Figure 4, the compounds were found to have inhibited metabolic activity in a time- and dose-dependent manner, and the highest efficiency was recorded in the case of the experimental group treated with compounds **1c** and **1d**.

The results obtained from the MTT assay were also used to determine IC_50_ values for each compound which are listed in Table 1. The IC_50_ values show that A549 cancer cells are more sensitive to the action of compounds **1c** and **1d** (IC_50_ = 27.04 and 21.22 × 10^−6^ M, respectively after 48 h) than to the other compounds from this series (IC_50_ > 50 × 10^−6^ M). Furthermore, these data corroborate the results obtained from the viability assay and the quantification of total cell number.

### 2.4. Quantification of Cell Number and Viability

The influence of the tacrine-coumarin compounds on total cell numbers was investigated after 24 h of treatment with the derivatives. As is shown in Figure 5, the total cell number decreased sharply (by more than 50%) in the case of cells treated with compounds **1c** and **1d**.

A simultaneous analysis of viability (Figure 5) showed that higher concentrations of compounds **1c** and **1d** had a weaker but nonetheless significant effect on cell survival. These results indicate that compounds **1c** and **1d** can influence total cell numbers and viability in a concentration-dependent manner.

### 2.5. Cell Cycle Distribution

The influence of the tacrine-coumarin hybrid molecules on the cell cycle distribution of cancer cells was investigated using flow cytometry. Data were collected from three independent experiments. As is shown in Table 2, the percentage of the cells at G_0_/G_1_ in the control group is 53.77 ± 1.43. The A549 cells were incubated with different concentrations of the studied compounds, and after 24 h incubation, the cells treated with compounds **1b** (at a higher concentration), **1c** and **1d** displayed an increased percentage of cells at the G_0_/G_1_ phase.

### 2.6. Clonogenic Assay

A549 cell lines were treated with two different concentrations of these derivatives. As is shown in Figure 6, no significant decrease in colony formation was observed, while a limited reduction was observed in the presence of a higher concentration of compound **1d**.

## 3. Discussion

DNA topoisomerases are crucial nuclear enzymes which control the topology of DNA by cleaving and re-joining the phosphodiester backbone of the DNA strand during various genetic processes. Clinical topoisomerase inhibitors act by generating topoisomerase-linked DNA breaks, blocking the religation of the cleavage complexes when a single drug molecule binds tightly at the interface of the topoisomerase-DNA cleavage complex [31]. As is well known, relaxed forms of supercoiled DNA migrate into a gel more slowly than non-relaxed DNA; this means that only the supercoiled band should be visible when topoisomerase activity is inhibited [32,33]. However, relaxation induced by *h*TOPI was inhibited when the concentration of the studied compounds was gradually increased, suggesting that the tacrine-coumarin compounds may cause a concentration-dependent inhibition of *h*TOPI. All of the studied compounds were found to have caused a complete inhibition at a concentration of 60 × 10^−6^ M. In order to evaluate whether the compounds can also inhibit topoisomerase IIα activity, the decatenation of catenated plasmid DNA was performed in the presence of the compounds. However, no significant inhibitory effect was detected, even at the highest concentration, and therefore we suggest that compounds **1a**–**2c** are unable to inhibit the topoisomerase IIα enzyme. It is important to understand that the cytotoxicity of topoisomerase inhibitors is due to the trapping of topoisomerase cleavage complexes, a process which should be distinguished from the associated topoisomerase catalytic inhibition. With the exception of molecularly defined settings, it is the topoisomerase cleavage complexes that kill the cancer cell [31]. TOPI plays an important role during the cell division process and we hypothesize that the inhibition of TOPI by tacrine-coumarin compounds can also influence cell division in A549 cell lines.

The presence of compounds **1a**–**2c** in A549 human adherent lung carcinoma cells was analysed by observing the fluorescence of the compounds in the green channel. As our results show, compound **1d** displayed the highest rate of detection in cells. In other samples, the fluorescence of derivatives was not distinguishable from the autofluorescence of the cancer cells. In cells, the compounds were distributed in the cytoplasm. Based on mitochondrial staining and the overall distribution of the signal, we were unable to confirm the accumulation of the compounds in the mitochondria or in other organelles or membranes. These observations suggest that no specific interaction through DNA binding is responsible for the observed cytotoxicity of these compounds. Flow cytometric analysis of the content of the derivatives present in the A549 cells revealed that compound **1c** was found to display the highest level of fluorescence.

The influence of the tacrine-coumarin compounds on total cell numbers was investigated after 24 h of treatment with the derivatives. The total cell number decreased sharply in the case of cells treated with compounds **1c** and **1d**. No significant changes were observed for cells treated with the other compounds from the series, but a simultaneous analysis of viability showed that higher concentrations of compounds **1c** and **1d** had a weaker but nonetheless significant effect on cell survival. These results indicate that compounds **1c** and **1d** can influence total cell numbers and viability in a concentration-dependent manner.

The ability of the studied compounds to inhibit the metabolic activity of A549 cancer cell lines was determined using an MTT assay. The compounds were found to have inhibited metabolic activity in a time- and dose-dependent manner, with the highest efficacy being recorded in the case of the experimental groups treated with compounds **1c** and **1d**. The IC_50_ values show that A549 cancer cells are more sensitive to the action of compounds **1c** and **1d** after 48 h than to the other compounds from this series. Furthermore, these data corroborate the results obtained from the viability assay and the quantification of total cell number. Tacrine was found to be a weak antiproliferative agent but we determined that the combination of tacrine and coumarin in a single molecule is more efficient against the cancer cell line.

Solarova et al. [34] have tested the cytotoxic and/or anti-cancer activities of tacrine-coumarin heterodimers **1a**–**2c** on 4T1 (mouse mammary carcinoma), MCF-7 (human breast adenocarcinoma), HCT116 (human colorectal carcinoma), A549 (human lung carcinoma), NMuMG (normal mouse mammary gland cells) and HUVEC (human endothelial cells isolated from umbilical vein) cell lines. Based on the obtained IC_50_ values, compounds **1a**–**2c** showed moderate to significant activity in the μM range. The A549 tumor cells proved to be the most resistant, with a proliferation not significantly different from the other cell lines after the administration of tacrine-coumarin derivatives **1b**–**1d**. Among the synthesized compounds, the tacrine-coumarin heterodimer with nine methylene groups between the two amino groups in the side chain exhibited the greatest efficacy. The authors proposed that tacrine-coumarin heterodimers **1a**–**2c** with longer side chains (replacing some methylene groups with amine moiety) had decreased the anticancer activity. The effect of the tacrine-coumarin hybrid compounds on the cancer cells is likely dependent on the length of the chain between the tacrine and coumarin moiety (compounds **1c**, **1d** = tacrine-(CH_2_)_8-9_-coumarin). However, when the -CH_2_ chain is interrupted by -NH groups, only a moderate inhibition effect on proliferation is recorded. Our attention was focused only on one A549 cancer cell line with the purpose of studying these compounds in more detail. According to our results, derivatives **1c** and **1d** displayed the best antiproliferative effect, a result which is similar to those reported by Solarova et al. The compounds with a greater length of chain between the tacrine and coumarin molecules showed an insignificant effect (**2a**–**2c**). As further evidence of the significance of hydrocarbon length, the antiproliferative activity increased in the order **1b** < **2b** < **1c** < **1d**.

A novel *bis*-tacrine and its congeners was tested for its potential as an anticancer agent by Hu et al. [35] An in-vitro cytotoxic evaluation of the compounds was carried out against a panel of 60 human cancer cell lines. Of the novel compounds, the butyl-linked *bis*-tacrine exhibited the strongest cytotoxic profile against non-small lung cancer cells. Congeners bearing a longer alkyl chain were on average 30- to 100-times less cytotoxic against these cancer cells.

We also investigated the influence of tacrine-coumarin hybrid molecules **1a**–**2c** on the cell cycle distribution of cancer cells. The A549 cells, which were treated with compounds **1b**–**1d**, displayed an increased percentage of cells at the G_0_/G_1_ phase. The data demonstrate that these compounds were also capable of inhibiting cells in the G_0_/G_1_ phase. These results are in agreement with those of Roldán-Pena et al. [36] who designed a series of tacrine-based homo- and heterodimer compounds incorporating an antioxidant tether which displayed antiproliferative activity. The compounds exhibited excellent in vitro antiproliferative activities against a panel of 6 human tumor cell lines, while cell cycle experiments indicated the accumulation of cells in the G_1_ phase of the cycle. A study by Janočková et al. [37] examined the effect of 7-MEOTA tacrine urea heterodimers on HL-60 cell lines and their results clearly demonstrated a significant accumulation of cells in the G_1_ phase.

In our study, the effects of derivatives **1c** and **1d** were found to be more prominent on the proliferation of cancer cells (demonstrated as a decline in total cell number) than on the viability of the cells. This agrees with the increased accumulation of cells in the G_0_/G_1_ phase. Finally, the inhibition of Topo I observed for the tacrine-coumarin compounds may also influence cell division in the A549 cell line. While all of these observed effects have a strong impact on the proliferation of cells (and consequently on the total cell number), this does not necessarily mean that the compounds also exert a cytotoxic effect (i.e., the compounds had not impaired cell viability to such a significant degree). Thus, the decreased cell number is primarily the result of inhibited proliferation rather than any cytotoxic effect of tacrine-coumarin hybrid compounds **1c** and **1d**.

In order to test the effect of the studied compounds on colony formation or clonogenic ability, we performed experiments with clonogenic assays. This is a simple technique which can identify biological alterations leading to irreversible losses of proliferative capacity and thus the loss of cells’ ability to form new colonies [38]. The changes were accompanied by a corresponding reduction in the percentage of cells in the S and G_2_/M phases. No significant decrease in colony formation was observed; a limited reduction was observed in the presence of a higher concentration of compound **1d**.

When we compare the results of all of the biological techniques used in this study, it is possible to suggest that the effect of the tacrine-coumarin hybrid compounds on cancer cells likely depends on the length of the chain between the tacrine and coumarin moiety. However, when the -CH_2_ chain is interrupted by -NH groups, only a moderate inhibition effect on proliferation is recorded.

## 4. Materials and Methods

### 4.1. Compounds

All chemicals and reagents were purchased from Sigma-Aldrich Chemie (Hamburg, Germany) and used without further purification. Human topoisomerase I- hTOPI, TOPOII (Inspiralis, Ltd., Norwich, UK), Ham Nutrient Mixture (Sigma-Aldrich, St. Louis, MO, USA) foetal bovine serum (Biosera, Boussens, France) and antibiotics (Antibiotic-Antimycotic 100 × and 50 × 10^−3^ g L^−1^ gentamicin; Biosera), MitoTrackerTM Red, DRAQ5TM, ProLongTM Gold Antifade Mountant (Thermo Fisher Scientific, Waltham, MA, USA). MTT (3-[4,5-dimethylthiazol-2-yl]-2,5-diphenyltetrazolium bromide) (Sigma-Aldrich, St. Louis, MO, USA) were used in the study.

The studied tacrine-coumarin hybrids (derivatives **1a**–**1d** and **2a**–**2c**) **1a**: *N*1-{6-[(1,2,3,4-tetra-hydroacridin-9-yl)amino]hexyl}-2-(7-hydroxy-2-oxo-2*H*-chromen-4-yl)acetamide, **1b**: *N*1-{7-[(1,2,3,4-tetrahydroacridin-9-yl)amino]heptyl}-2-(7-hydroxy-2-oxo-2*H*-chromen-4-yl)acetamide, 1c: *N*1-{8-[(1,2,3,4-tetrahydroacridin-9-yl)amino]octyl}-2-(7-hydroxy-2-oxo-2*H*-chromen-4-yl)acetamide, 1d: *N*1-{9-[(1,2,3,4-tetrahydroacridin-9-yl) amino]nonyl}-2-(7-hydroxy-2-oxo-2*H*-chromen-4-yl)acetamide, **2a**: 2-(7-hydroxy-2-oxo-2*H*-chromen-4-yl)-*N*-[6-(1,2,3,4-tetrahydroacridin-9-ylamino)hexyl]acetamide, **2b:**
*N*1-[3-({3-[(1,2,3,4-tetrahydroacridin-9-yl)amino]propyl}amino)propyl]-2-(7-hydroxy-2-oxo-2*H*-chromen-4-yl)acetamide, **2c:** 2-(7-hydroxy-2-oxo-2*H*-chromen-4-yl)-*N*-[3-[2-[3-(1,2,3,4-tetrahydroacridin-9-ylamino)propylamino]ethylamino]propyl]acetamide (Figure 7) [1] were dissolved in dimethyl sulfoxide (DMSO, Fluka) to a final concentration of 5 × 10^−2^ M.

### 4.2. Topoisomerase Relaxation Assay

The effects of compounds **1a**–**2c** on the relaxation of plasmid DNA with human topoisomerase I (*h*TOPI) were investigated using negatively supercoiled plasmid *p*BR322 (0.5 × 10^−6^ g) incubated for 30 min at 37 °C with 2 units of *h*TOPI (Inspiralis, Ltd., Norwich, UK) in both the presence and absence of the studied tacrine-coumarin hybrid molecules at concentrations of 5, 30 and 60 × 10^−6^ M, respectively. The method used to perform the experiment of TOPOII has been published previously [37].

### 4.3. Cell Culture

Human lung carcinoma cell lines A549 were purchased from the American Type Culture Collection (ATCC, Rockville, MD, USA). The cells were incubated in Kaighn’s modification of F-12 Ham Nutrient Mixture supplemented with 10% fetal bovine serum (FBS) and antibiotics (1% Antibiotic-Antimycotic 100 × and 50 × 10^−3^ g L^−1^ gentamicin; Biosera) at 37 °C, 95% humidity and 5% CO_2_. The cells (10,000/cm^−2^) were seeded on 12-well μ-Chamber slides (ibidi GmbH, Martinsried, Germany) on 6, 12 and 96-well plates (TPP, Trasadingen, Switzerland) and left to settle for 24 h. This incubation method has been published previously [39].

### 4.4. Intracellular Localization and Cytotoxicity Assays

The derivatives were visualized in cells with an Argon Laser at 488 nm and fluorescence was captured at a range of 510–560 nm with identical exposure parameters used for all samples. Microphotographs were taken with a 100 × oil lens and were then captured and analysed using LAS AF software (Leica Microsystems, Mannheim, Germany).

Floating and adherent cells were harvested both 6 and 24 h after treatment with the derivatives, washed in PBS and resuspended in Hank’s balanced salt solution (HBSS). Intracellular levels of derivatives were detected using a BD FACSCalibur flow cytometer (Becton Dickinson, San Jose, CA, USA) and determined based on fluorescence excitation at 488 nm. Fluorescence was detected via a 530/30 nm band-pass filter (FL-1), 585/42 band-pass filter (FL-2) and 670 nm long-pass filter (FL-3). The results were analyzed using FlowJo software (TreeStar Inc., Ashland, OR, USA).

MTT assays were added to the cells in a 96-well plate (at a final concentration of 0.5 g L^−1^) 24 and 48 h after treatment with the derivatives [39]. The absorbance (λ = 584 nm) was measured using a BMG FLUOstar Optima (BMG Labtechnologies GmbH, Offenburg, Germany). The results were evaluated as percentages of the absorbance of the untreated control. IC_50_ values for each derivative were extrapolated from a sigmoidal fit to the metabolic activity data using OriginPro 8.5.0 SR1 (OriginLab Corp., Northampton, MA, USA).

For an assessment of total cell numbers and viability within individual experimental groups, floating and adherent cells were harvested 24 h after treatment with the studied derivatives and evaluated using a Bürker chamber (Paul Marienfeld GmbH&Co.KG, Lauda-Königshofen) with eosin staining. The total cell number was expressed as a percentage of the untreated control of the total cell number. Viability was expressed as a percentage of viable, eosin negative cells.

Details of the experiment with flow cytometric analysis have been published previously [38]. The DNA content was analysed using a BD FACSCalibur flow cytometer (Becton Dickinson) with a 488 nm argon-ion excitation laser, and fluorescence was detected via a 585/42 nm band-pass filter (FL-2). ModFit 3.0 software (Verity Software House, Topsham, ME, USA) was used to generate DNA content frequency histograms and to quantify the percentage of cells in the individual cell cycle phases.

### 4.5. Clonogenic Assay

The cells were counted using a Bürker chamber with eosin staining and 800 viable cells per well were seeded in 6-well plates. After 7 days of incubation under standard conditions, the cells in the plates were fixed and stained with 1% methylene blue dye in methanol. Visualized colonies were scanned, counted and the results were evaluated as percentages of the untreated control.

### 4.6. Statistical Analysis

Data were analyzed using a one-way ANOVA with Tukey´s post-test and are expressed as the mean ± standard deviation (SD) of at least three independent experiments. The experimental groups treated with the derivatives were compared with the control group: * *p* < 0.05.

## 5. Conclusions

This study has investigated a series of novel derivatives with both tacrine and coumarin pharmacophores, compounds **1a**–**2c**. Our results suggest that the novel derivatives had completely inhibited topoisomerase activity at a concentration of 60 × 10^−6^ M. The presence and content of the novel tacrine-coumarin hybrid molecules after introduction to A549 human adherent lung carcinoma cell lines were also investigated using confocal microscopy. Only compound **1d** was found to be present in the cell lines to a substantial degree. The IC_50_ values which were determined in this assay show that A549 cancer cell lines are more sensitive to the effect of compounds **1c** and **1d** (IC_50_ = 27.04 and 21.22 × 10^−6^ M, respectively after 48 h) than to the other compounds in the series. A simultaneous analysis of viability showed that higher concentrations of compounds **1c** and **1d** had a weaker but nonetheless significant effect on cell survival. These results indicate that compounds **1c** and **1d** are capable of influencing total cell numbers and viability in a concentration-dependent manner. The findings presented in this paper suggest that these tacrine-coumarin molecules exhibit promising potential as topoisomerase I inhibitors with anticancer activity against A549 human adherent lung carcinoma cells in addition to their well-known anticholinesterase effects [1] and may also serve as BSA-interacting agents [40]. These features would be of considerable use in the development of drugs with enhanced or more selective effects and greater clinical efficacy.

## Figures and Tables

**Figure 1 molecules-26-01133-f001:**
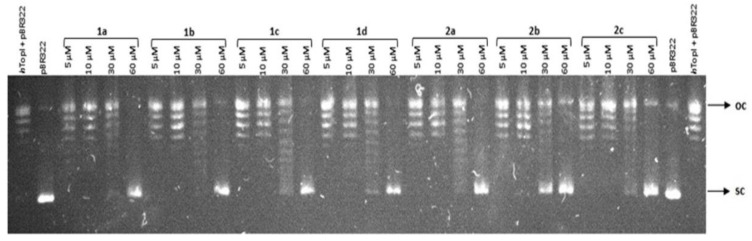
Electrophoresis agarose gel showing inhibitory effects of tacrine-coumarin compounds **1a**–**2c** on human topoisomerase I (*h*TOPI) activity. Supercoiled plasmid DNA (*p*BR322—negative control) was incubated for 30 min with 2 U of *h*TOPI in the absence (lines *h*TOPI + *p*BR322—positive control) and presence of varying concentrations of compounds **1a-2c** (lines **1a**–**2c**).

**Figure 2 molecules-26-01133-f002:**
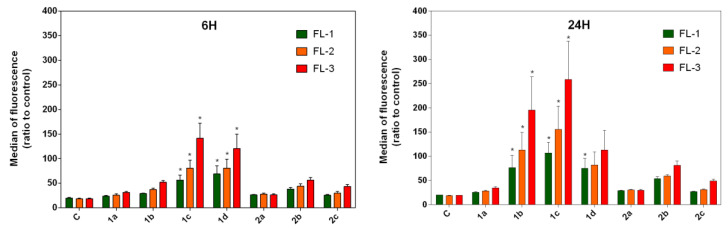
Flow cytometric analysis of intracellular level of derivatives **1a**–**2c**. Intrinsic fluorescence of compounds was detected after excitation at 488 nm and the emission was measured using a 530/30 nm band-pass filter (FL-1), 585/42 band-pass filter (FL-2) and 670 nm long-pass filter (FL-3). The results are presented as the mean values ± SD of three independent experiments; statistical significance * *p* < 0.05 for each experimental group compared to the untreated control.

**Figure 3 molecules-26-01133-f003:**
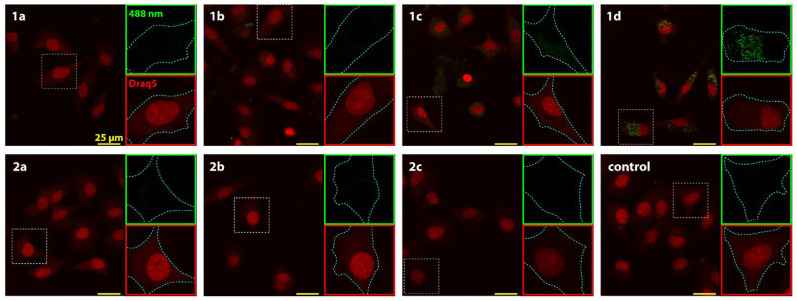
Confocal microscopy images of A549 cancer cell lines after 24 h incubation with compounds **1a**–**2c**. The microphotographs show the representative images of the samples with merged channels. Compounds **1a**–**2c** were visualized in cells with a 488 nm laser and the fluorescence was captured at the range of 510–560 nm (green insets). Red insets show nuclear labelling with Draq5. Scale bar = 25 μm.

**Figure 4 molecules-26-01133-f004:**
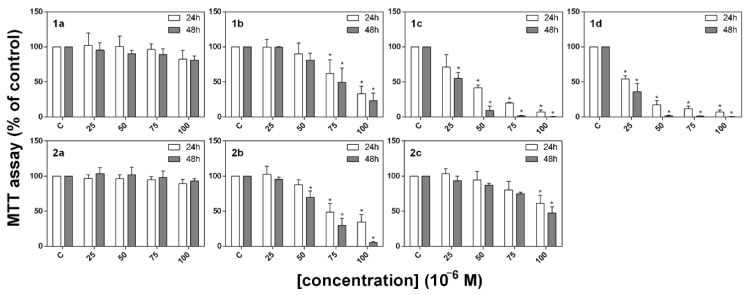
Effect of tacrine-coumarin hybrid compounds **1a**–**2c** on metabolic activity evaluated by MTT assay in A549 cancer cell lines. MTT assays are expressed as percentages of the untreated control. The results are presented as the mean values ± SD of three independent experiments; statistical significance (*): *p* < 0.05 for each experimental group compared to the untreated control.

**Figure 5 molecules-26-01133-f005:**
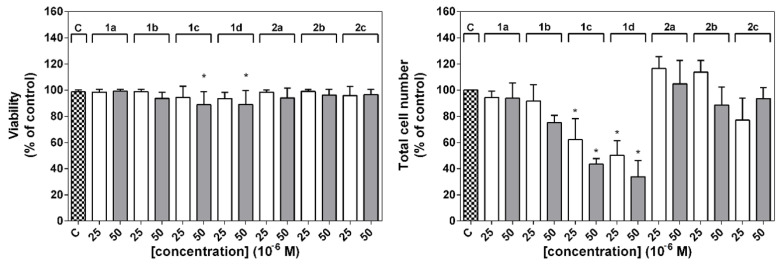
Effect of tacrine-coumarin hybrid compounds **1a**–**2c** on viability and total cell numbers in A549 cancer cell lines. The viability and total cell number were evaluated 24 h after the addition of the derivatives and are expressed as a percentage of the viable, eosin negative cells or as a percentage of the untreated control of the total cell number, respectively. The results are presented as the mean values ± SD of three independent experiments; statistical significance * *p* < 0.05 for each experimental group is compared to the untreated control.

**Figure 6 molecules-26-01133-f006:**
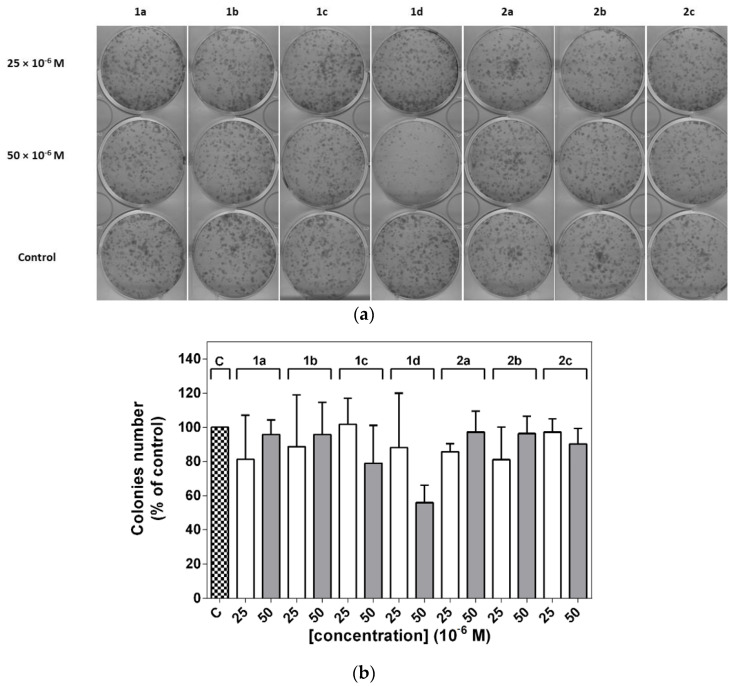
Clonogenic assay of A549 cancer cell lines. Cells were untreated (control) or treated with different concentrations of tacrine-coumarin hybrid derivatives **1a**–**2c** for 24 h. (**a**) The experimental and (**b**) graphical presentation of the results. The results of the subsequent 7-day cultivation are presented as the mean values ± SD of three independent experiments.

**Figure 7 molecules-26-01133-f007:**
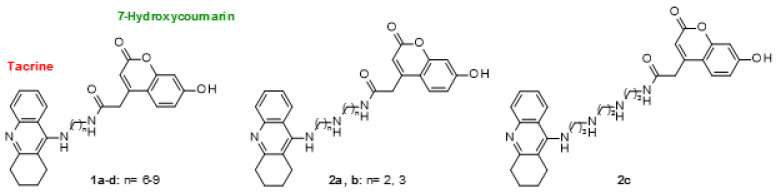
Structure of tacrine-coumarin hybrid molecules (derivatives **1a**–**1d** and **2a**–**2c**) [1].

**Table 1 molecules-26-01133-t001:** IC_50_ values of tacrine-coumarin hybrid molecules **1a–2c** in A549 cancer cell lines.

Compound	^a^ IC_50_ (× 10^−6^ M)	
	24 h	48 h
**1a**	n.d.	n.d.
**1b**	83.54	74.27
**1c**	42.36	27.04
**1d**	27.25	21.22
**2a**	n.d.	n.d.
**2b**	74.05	62.33
**2c**	n.d.	98.68

n.d.—not detected, ^a^ IC_50_—the concentration of the compound at which 50% of metabolic activity is inhibited.

**Table 2 molecules-26-01133-t002:** Effect of tacrine-coumarin hybrid compounds **1a**–**2c** on cell cycle distribution.

Compound	Concentration (× 10^−6^ M)	G_1_	S	G_2_
Control	0	53.77 ± 1.43	33.43 ± 0.71	12.81 ± 0.94
**1a**	25	57.03 ± 0.86	31.02 ± 0.65	11.94 ± 1.37
	50	57.88 ± 1.68	30.56 ± 0.72	11.56 ± 0.99
**1b**	25	59.08 ± 1.32	29.64 ± 0.22	11.29 ± 1.36
	50	71.13 ± 1.59 *	21.72 ± 1.22 *	7.15 ± 1.29 *
**1c**	25	79.97 ± 1.25 *	14.64 ± 1.56 *	5.39 ± 0.9 *
	50	91.86 ± 2.24 *	6.09 ± 1.98 *	2.05 ± 0.62*
**1d**	25	86.21 ± 1.67 *	10.73 ± 0.97 *	3.06 ± 0.85 *
	50	80.85 ± 2.30 *	12.54 ± 3.46 *	6.61 ± 1.84 *
**2a**	25	55.54 ± 1.43	33.00 ± 1.03	11.46 ± 1.41
	50	56.94 ±1.20	32.06 ± 0.22	11.01 ± 1.07
**2b**	25	58.18 ± 0.35	30.51 ± 1.20	11.30 ± 0.86
	50	56.69 ± 2.40	32.49 ± 1.56	10.83 ± 0.9
**2c**	25	57.03 ± 2.74	31.77 ± 1.73	11.20 ± 1.21
	50	57.68 ± 1.24	31.80 ± 0.6	10.51 ± 1.67

* Statistical significance: *p* < 0.05 for each experimental group compared to untreated control.

## Data Availability

The data presented in this study are available on request from the corresponding author.

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
