# Peer review of "Tacrine-Coumarin Derivatives as Topoisomerase Inhibitors with Antitumor Effects on A549 Human Lung Carcinoma Cancer Cell Lines"

_molecules, 2021, doi:10.3390/molecules26041133_

Round 1
Reviewer 1 Report
The manuscript described the antiproliferative and topoisomerase in vitro activities of tacrine-coumarin hybrid compounds 1a-2c bonded by different long alkyl chains, which contain either -CH2 groups or -NH group. This compounds were primarily investigated in the study "Targeting copper (II)-induced oxidative stress and the acetylcholinesterase systemin Alzheimer's disease using multifunctional tacrine-coumarin hybrid molecules" (J.Inorg. Biochem. 2016, (161), 52-62), where the synthesis of the given compounds is described and the authors describe this fact also in the presented manuscript with reference to the original literature. In my opinion, it is very interesting to look at the use of these compounds as topoisomerase inhibitors with antitumor effect since tacrine or its derivatives are characterized by inhibitory properties against topoisomerase I and II.
However, the manuscript suffers from many imprecisions, some of them of high significance and therefore could accepted by the journal only after the major revisions. The suggestions are listed below:
- Page 8 lines 248-252: Please correct the compound names: 2a to N1-[2-({2-[(1,2,3,4-tetrahydroacridin-9-yl)amino]ethyl}amino)ethyl]-2-(7-hydroxy-2-oxo-2H-chromen-4-yl)acetamide; and 2c to N1-[3-{[2-({3-[(1,2,3,4-tetrahydroacridin-9-yl)amino]propyl}amino)ethyl]amino}propyl]-2-(7-hydroxy-2-oxo-2H-chromen-4-yl)acetamide
- Please indicate in the present manuscript the structures of compounds 1a-2c.
- Page 5: Figure 5 shows effect of tacrine-coumarin hybrid compounds 1a-2c on viability and total cell numbers in A549 cancer cell line. It would be appropriate to broaden the discussion on these results. Plus, the total number of cells decreased sharply (more than 50%) in the case of cells treated with compounds 1c and 1d, although the analysis of viability showed that higher concentrations of compounds 1c and 1d had a weaker effect on the cell survival.
- In my opinion, the discussed results of derivatives 1a-2c in the present manuscript should be compared with the data of tacrine or its published derivatives either experimentally or with the described results in literature. Based on these results, then describe whether the introduction of the coumarin part has improved or worsened the discussed results from the current manuscript.
- The assertion "When we compare the results of all of the biological techniques used in this study, it is possible to suggest that the effect of the tacrine-coumarin hybrid compounds on cancer cells probably depends on the length of the chain between the tacrine and coumarin moiety (the longer chain, the higher cytotoxicity). However, when the -CH2 chain is interrupted by -NH groups, only a moderate inhibition effect on proliferation is recorded" should be based on experimental logP values for compounds 1a-2c, and based on these results, the possibility of increased cytotoxicity and further penetration of the studied compounds 1a-2c into cells associated with increased lipophilicity of the studied compounds should be discussed.
Author Response
Reviewer 1
Dear reviewer,
At the beginning we would like to thank you for all the valuable comments suggested by the opponents. We went through all the questions step by step and answered them. All changes in the text are marked in yellow. We hope we were able to answer all the questions in detail.
Page 8 lines 248-252: Please correct the compound names: 2a to N1-[2-({2-[(1,2,3,4-tetrahydroacridin-9-yl)amino]ethyl}amino)ethyl]-2-(7-hydroxy-2-oxo-2H-chromen-4-yl)acetamide; and 2c to N1-[3-{[2-({3-[(1,2,3,4-tetrahydroacridin-9-yl)amino]propyl}amino) ethyl]amino}propyl]-2-(7-hydroxy-2-oxo-2H-chromen-4-yl)acetamide –
Please indicate in the present manuscript the structures of compounds 1a-2c.
We would like to apologize for the inaccurate description. We have corrected the names of the compounds and have inserted a description of the structures of the compounds.
- Page 5: Figure 5 shows effect of tacrine-coumarin hybrid compounds 1a-2c on viability and total cell numbers in A549 cancer cell line. It would be appropriate to broaden the discussion on these results. Plus, the total number of cells decreased sharply (more than 50%) in the case of cells treated with compounds 1c and 1d, although the analysis of viability showed that higher concentrations of compounds 1c and 1d had a weaker effect on the cell survival.
The effects of compounds 1c and 1d were in fact more prominent on the proliferation of cancer cells (demonstrated as a decline in total cell number) than on the viability of the cells. This agrees with the increased accumulation of cells in the G0/G1 phase. Finally, the inhibition of Topo I observed with tacrine-coumarin compounds can also influence cell division in the A549 cell line. While all of these observed effects have a strong effect on the proliferation (and consequently on the total cell number), this does not necessarily mean that they are also cytotoxic (i.e. the compounds were not seen to have impaired cell viability to such a significant degree). Thus, the decreased cell number is primarily the result of inhibited proliferation rather than the cytotoxic action of tacrine-coumarin hybrid compounds 1c and 1d. We have included this text in the manuscript.
- In my opinion, the discussed results of derivatives 1a-2c in the present manuscript should be compared with the data of tacrine or its published derivatives either experimentally or with the described results in literature. Based on these results, then describe whether the introduction of the coumarin part has improved or worsened the discussed results from the current manuscript.
This has been discussed in the Discussion section.
- The assertion "When we compare the results of all of the biological techniques used in this study, it is possible to suggest that the effect of the tacrine-coumarin hybrid compounds on cancer cells probably depends on the length of the chain between the tacrine and coumarin moiety (the longer chain, the higher cytotoxicity). However, when the -CH2 chain is interrupted by -NH groups, only a moderate inhibition effect on proliferation is recorded" should be based on experimental logP values for compounds 1a-2c, and based on these results, the possibility of increased cytotoxicity and further penetration of the studied compounds 1a-2c into cells associated with increased lipophilicity of the studied compounds should be discussed.
Thank you for your suggestion. We did not resolve the issue of penetration in this way, as we were confident with the confocal microscopy results showing that the substances had penetrated through the membrane.
Reviewer 2 Report
In this manuscript, Konkolová et al. demonstrated the role of seven new compounds having both tacrine and coumarin pharmacophores to inhibit topoisomerase I activity and cancer cell growth in-vitro.
There are a few interesting findings, but the manuscript needs improvement based on these comments for becoming worthy of getting published in 'Molecules'
1. The manuscript is written like a report or set of observations rather than a scientific paper where the implications and the consequences should be clearly mentioned and discussed. The style should be improved.
2. There are a few grammatical mistakes here and there e.g. line 72, 'is shows...' and 170 'As is well known...'
3. In the literature review, while discussing lung cancer therapy, the authors mentioned radiotherapy and chemotherapy as the method of intervention. However, we now have some very good FDA-approved targeted therapy agents and chemotherapy. This part about lung cancer therapies also lacks the reference. A couple of more lines are needed here.
4. Importantly, why the authors did not mention in the references, one of the papers from Acta Chim. Slov. 2018 by Solárová et al., where they tested Tacrine-Coumarin derivatives in many cell lines including A549? This paper's reference should be given, and results from this paper should be discussed with their own results. One of the co-authors is also mentioned in this 2018 paper.
5. In the discussion, the authors discussed the putative reason for compound 1d's effectiveness as '...length of the chain between the tacrine and coumarin moi-230 ety (the longer chain, the higher cytotoxicity)...'. In the manuscript, though, they did not draw the structures of any of the compounds. Structures are needed to be drawn if they compare the properties of these based on their chemical structure based on the last lines in the discussion.
6. For Fig. 1, It is not clearly mentioned what is here 'positive control' and the 'negative control'.
7. Fig. 2 is not explained well. What is FL-1, FL-2, and FL-3? Are these the intrinsic fluorescence of the compounds? What is the significance of this fluorescence?
8. Fig 3 mentions 'stained with MitoTrackerTM Red and DRAQ5TM, respectively. Green dots represent the studied compounds scattered freely in the cytoplasm.' In the figure, though, there is only the red channel and green. How to differentiate between MitoTracker and DRAQ5? Is red in the pictures, MitoTracker or DRAQ5? Scale bar absent. What is the concentration of drugs used in this experiment? Indicate the green cells with arrows and zoom them in the inset.
9.Why the concentrations used in Fig. 4 not used in Fig. 1 and vice-versa. Inconsistency in the concentrations of the drugs used in different experiments makes the results difficult to interpret at the mechanistic and functional level.
10. For Fig 5, to me, based on the overlapping error bars, the results do not seem to be significant, especially for 5A, compounds 1c and 1d. Please provide the exact p-values to the reviewers for confirmation. Similar looking error bars in Fig 4 are insignificant, which is confusing.
11. For line 148-149 and 217-220, this result indicates and signifies what?
12. For fig. 6, Manual colony counting is outdated and highly imprecise due to touching colonies. Use automated tools like in this paper to redo the analysis. https://pubmed.ncbi.nlm.nih.gov/24647355/
14. Neither the limitations nor the future directions for this study are discussed in the discussion.
15. In general, the compounds' effective concentration is very high, more than 50uM, which cannot be used in any other more physiological system apart from the cell lines. Tweaking of the groups e.g. for 1d will be needed for future investigations. This all should be openly discussed in the discussion. Comparison with other similar studies is needed in the discussion.
16. Did the authors try any other cell line for these compounds? Though not necessary for the scope of this manuscript but interesting to know.
Author Response
Reviewer 2
Dear reviewer,
At the beginning we would like to thank you for all the valuable comments suggested by the opponents. We went through all the questions step by step and answered them. All changes in the text are marked in yellow. We hope we were able to answer all the questions in detail.
- The manuscript is written like a report or set of observations rather than a scientific paper where the implications and the consequences should be clearly mentioned and discussed. The style should be improved.
We have tried our best in the respect, and all of the changes we have made are marked in yellow in the text.
- There are a few grammatical mistakes here and there e.g. line 72, 'is shows...' and 170 'As is well known...'
We apologize for the typing errors; we have corrected them.
- In the literature review, while discussing lung cancer therapy, the authors mentioned radiotherapy and chemotherapy as the method of intervention. However, we now have some very good FDA-approved targeted therapy agents and chemotherapy. This part about lung cancer therapies also lacks the reference. A couple of more lines are needed here.
We inserted the mentioned text into the Introduction section and supplemented the literature related to this issue.
- Importantly, why the authors did not mention in the references, one of the papers from Acta Chim. Slov. 2018 by Solárová et al., where they tested Tacrine-Coumarin derivatives in many cell lines including A549? This paper's reference should be given, and results from this paper should be discussed with their own results. One of the co-authors is also mentioned in this 2018 paper.
We have included the following into the text: Solarova et al. [34] have tested the cytotoxic and/or anti-cancer activities of tacrine-coumarin heterodimers 1a-2c on 4T1 (mouse mammary carcinoma), MCF-7 (human breast adenocarcinoma), HCT116 (human colorectal carcinoma), A549 (human lung carcinoma), NMuMG (normal mouse mammary gland cells), and HUVEC (human endothelial cells isolated from umbilical vein) cell lines. Based on the obtained IC50 values, compounds 1a-2c showed moderate to significant activity in the μM range. The A549 tumor cells proved to be the most resistant, with a proliferation not significantly different from the other cell lines after the administration of tacrine-coumarin derivatives 1b-1d. Among the synthesized compounds, the tacrine-coumarin heterodimer with nine methylene groups between the two amino groups in the side chain exhibited the greatest efficacy. The authors proposed that tacrine-coumarin heterodimers 1a-2c with longer side chains (replacing some of the methylene groups with amine moiety) had decreased the anticancer activity.
- In the discussion, the authors discussed the putative reason for compound 1d's effectiveness as '...length of the chain between the tacrine and coumarin moi-230 ety (the longer chain, the higher cytotoxicity)...'. In the manuscript, though, they did not draw the structures of any of the compounds. Structures are needed to be drawn if they compare the properties of these based on their chemical structure based on the last lines in the discussion.
We have inserted this into the text.
- For Fig. 1, It is not clearly mentioned what is here 'positive control' and the 'negative control'.
Thank you for your suggestion. The hTopI + pBR322 line (a complete reaction mixture with enzyme and plasmid DNA) was the positive control, and the negative control was the pBR322 line (a reaction mixture without enzyme, but with pBR322 plasmid DNA).
- Fig. 2 is not explained well. What is FL-1, FL-2, and FL-3? Are these the intrinsic fluorescence of the compounds? What is the significance of this fluorescence? –
This refers to the flow cytometric analysis of the intracellular level of derivatives 1a-2c. The intrinsic fluorescence of the compounds was detected after excitation at 488 nm and the emission was measured with a 530/30 nm band-pass filter (FL-1), 585/42 band-pass filter (FL-2) and 670 nm long-pass filter (FL-3). The results are presented as the mean values ± SD of three independent experiments; statistical significance (*): p < 0.05 for each experimental group compared to the untreated control.
This analysis was performed in order to detect the accumulation of the compounds within the cells by exploiting their natural fluorescence. This allowed us to correlate their accumulation with the observed effects on the cellular parameters. The accumulation of the compounds was then investigated in more detail with respect to their specific intracellular distribution using confocal microscopy.
- Fig 3 mentions 'stained with MitoTrackerTM Red and DRAQ5TM, respectively. Green dots represent the studied compounds scattered freely in the cytoplasm.' In the figure, though, there is only the red channel and green. How to differentiate between MitoTracker and DRAQ5? Is red in the pictures, MitoTracker or DRAQ5? Scale bar absent. What is the concentration of drugs used in this experiment? Indicate the green cells with arrows and zoom them in the inset.
We would like to apologize for the inaccurate description of the microphotographs in the figure legend. Indeed, we performed several stainings to visualize the chemical compounds and also the cellular organelles, including the mitochondria. However, during the preparation of the manuscript, we finally decided to show only the microphotographs with the chemical compounds excited at 488 nm and the cell nuclei stained with Draq5 (excited at 643 nm) in order to keep the manuscript straightforward and easy to follow. In line with this intention, we have reworked both the figure and the figure legend. The insets now show the cells in both channels and in more detail. The scale bars were also added in line with your suggestions.
- Why the concentrations used in Fig. 4 not used in Fig. 1 and vice-versa. Inconsistency in the concentrations of the drugs used in different experiments makes the results difficult to interpret at the mechanistic and functional level.
Thank you for your suggestion. The inconsistency in the concentration of the drugs used in the experiments originated from the different planning of the various experiments. The influence of topoisomerase I was measured in various dilution series, and the data in Fig. 1 represent a selection of values from these dilution series. However, the concentrations used in the experiments on the cells were determined based on the first measurement performed in a single comprehensive/extended dilution series from 5 µM to 100 µM.
- For Fig 5, to me, based on the overlapping error bars, the results do not seem to be significant, especially for 5A, compounds 1c and 1d. Please provide the exact p-values to the reviewers for confirmation. Similar looking error bars in Fig 4 are insignificant, which is confusing.
Statistical analysis was performed using GraphPad Prism 5 software and the data was analyzed using a one-way ANOVA with Tukey´s multiple comparison test. The reviewer can verify the statistical significance of the results for Fig. 5 in the export data table which is presented below.
|
Table Analyzed |
24 h |
|
|
|
|
|
|
|
|
|
|
|
|
One-way analysis of variance |
|
|
|
|
|
|
P value |
0,0036 |
|
|
|
|
|
P value summary |
** |
|
|
|
|
|
Are means signif. different? (P < 0.05) |
Yes |
|
|
|
|
|
Number of groups |
15 |
|
|
|
|
|
F |
2,857 |
|
|
|
|
|
R squared |
0,4598 |
|
|
|
|
|
|
|
|
|
|
|
|
ANOVA Table |
SS |
df |
MS |
|
|
|
Treatment (between columns) |
944,6 |
14 |
67,47 |
|
|
|
Residual (within columns) |
1110 |
47 |
23,61 |
|
|
|
Total |
2054 |
61 |
|
|
|
|
|
|
|
|
|
|
|
Tukey's Multiple Comparison Test |
Mean Diff. |
q |
Significant? P < 0.05? |
Summary |
95% CI of diff |
|
C vs 1a25 |
1,925 |
0,8679 |
No |
ns |
-9.318 to 13.17 |
|
C vs 1a50 |
1,250 |
0,5636 |
No |
ns |
-9.993 to 12.49 |
|
C vs 1b25 |
1,375 |
0,6200 |
No |
ns |
-9.868 to 12.62 |
|
C vs 1b50 |
3,275 |
1,477 |
No |
ns |
-7.968 to 14.52 |
|
C vs 1c25 |
2,083 |
0,8575 |
No |
ns |
-10.23 to 14.40 |
|
C vs 1c50 |
12,88 |
5,805 |
Yes |
* |
1.632 to 24.12 |
|
C vs 1d25 |
6,430 |
3,090 |
No |
ns |
-4.117 to 16.98 |
|
C vs 1d50 |
11,31 |
5,436 |
Yes |
* |
0.7630 to 21.86 |
|
C vs 2a25 |
0,7750 |
0,3494 |
No |
ns |
-10.47 to 12.02 |
|
C vs 2a50 |
7,375 |
3,325 |
No |
ns |
-3.868 to 18.62 |
|
C vs 2b25 |
0,3167 |
0,1303 |
No |
ns |
-12.00 to 12.63 |
|
C vs 2b50 |
2,050 |
0,9243 |
No |
ns |
-9.193 to 13.29 |
|
C vs 2c25 |
5,300 |
2,390 |
No |
ns |
-5.943 to 16.54 |
|
C vs 2c50 |
4,325 |
1,950 |
No |
ns |
-6.918 to 15.57 |
- For line 148-149 and 217-220, this result indicates and signifies what?
We apologize for the inaccurate description; we have corrected them.
- For fig. 6, Manual colony counting is outdated and highly imprecise due to touching colonies. Use automated tools like in this paper to redo the analysis. https://pubmed.ncbi.nlm.nih.gov/24647355/
Thank you for your suggestion. In the presented study, we used Clono-Counter software (https://www.ncbi.nlm.nih.gov/pmc/articles/PMC1770926/) for colony counting, but the number of colonies was also manually adjusted due to the natural spread of colonies formed by the A549 lines with their low contrast even after staining.
- Neither the limitations nor the future directions for this study are discussed in the discussion.
This has been discussed in the Discussion section.
- In general, the compounds' effective concentration is very high, more than 50uM, which cannot be used in any other more physiological system apart from the cell lines. Tweaking of the groups e.g. for 1d will be needed for future investigations. This all should be openly discussed in the discussion. Comparison with other similar studies is needed in the discussion.
The aim of this work was to show that these structurally novel tacrine-coumarin derivatives may also exhibit anticancer properties in addition to their well-known anticholinesterase effects. The main goals of the work were to prove the multifunctional properties of tacrine congeners 1a-2c and to examine the antiproliferative and topoisomerase activities of these derivatives in more detail.
- Did the authors try any other cell line for these compounds? Though not necessary for the scope of this manuscript but interesting to know.
We decided not to test the compounds against other cell lines because our colleagues have already tested Tacrine-Coumarin derivatives on many cancer cell lines (Solárova et al.).
Round 2
Reviewer 1 Report
I thank the authors for improving interpretation of the results.
I have only a minor comment:
- Page 9 lines 306-307: Please indicate the designation of compounds 1c and 1d "bold".
- Page 9 line 308: Please correct the compound name: 2a to N1-[2-({2-[(1,2,3,4-tetrahydroacridin-9-yl)amino]ethyl}amino)ethyl]-2-(7-hydroxy-2-oxo-2H-chromen-4-yl)acetamide. The name given in the manuscript is not correct for the compound designated 2a.
- The literature references is incorrect in the text and in the list of references: e.g. Solarova et al., Hu et al., Roldán-Pena et al., Janočková et al. Literature 40 is not listed in the list of references. Please check the designation of references in the submitted manuscript.
Reviewer 2 Report
The authors managed to address well the major scientific concerns and can now be considered for publication!